# Sex-Related Differences in Glioblastoma: A Single-Center Retrospective Cohort Study

**DOI:** 10.3390/biomedicines13071715

**Published:** 2025-07-14

**Authors:** Chiara Prosperetti, Meltem Yenigün, Alberto Pagnamenta, Payam Tabaee Damavandi, Giulio Disanto, Francesco Marchi, Vittoria Espeli, Barbara Muoio, Paolo Spina, Gianfranco Pesce, Pamela Agazzi

**Affiliations:** 1Neurocenter of Southern Switzerland, Ente Ospedaliero Cantonale, 6900 Lugano, Switzerland; payam.tabaeedamavandi@eoc.ch (P.T.D.); giulio.disanto@eoc.ch (G.D.); francesco.marchi@eoc.ch (F.M.);; 2Faculty of Biomedical Sciences, Università della Svizzera Italiana (USI), 6900 Lugano, Switzerland; 3Support Area, Ente Ospedaliero Cantonale, 6500 Bellinzona, Switzerland; meltem.yeniguen@eoc.ch; 4Clinical Trial Unit, Ente Ospedaliero Cantonale, 6900 Lugano, Switzerland; alberto.pagnamenta@eoc.ch; 5Oncology Institute of Southern Switzerland, Ente Ospedaliero Cantonale, 6500 Bellinzona, Switzerlandbarbara.muoio@eoc.ch (B.M.); gianfrancoangelo.pesce@eoc.ch (G.P.); 6Institute of Pathology, Ente Ospedaliero Cantonale, 6601 Locarno, Switzerland; paolo.spina@eoc.ch

**Keywords:** glioblastoma and sex differences, sex disparities in brain cancer, glioblastoma epidemiology, precision medicine and brain cancer, molecular biomarkers in glioma

## Abstract

**Background**: Sex differences play a significant role in the epidemiology, biology, and outcomes of many cancers, including glioblastoma (GB), the most common and aggressive primary brain tumor. GB is more frequent in males, while females tend to have longer survival, though the underlying reasons for these differences remain poorly understood. Potential contributors include hormonal influences, sex-specific risk factors, and treatment disparities. Understanding these differences is critical for optimizing personalized treatment strategies. **Methods**: We conducted a retrospective analysis of patients with gliomas from a neuro-oncological database, with a primary focus on GB cases. Variables collected included sex, age, tumor type, molecular biomarker, and treatment modalities. The primary objective was to assess sex-based differences in tumor characteristics and outcomes, while the secondary objective was to identify predictors of time to progression and mortality. **Results**: The cohort comprised 125 GB, 48 astrocytomas, and 16 oligodendrogliomas, with no significant sex-based differences in age or tumor type distribution. Among GB patients, multifocality was more prevalent in females (14% vs. 8%; *p* = 0.01); also, EGFR amplification was more frequent in females (25.5% vs. 52.5%; *p* = 0.007). Males received chemotherapy (80% vs. 63%; *p* = 0.04) and radiotherapy (84% vs. 67%; *p* = 0.03) more frequently than females. Survival was positively associated with *MGMT* methylation (*p* = 0.002) and negatively associated with TERT mutation (*p* = 0.01). Multivariable analysis identified TERT mutation as a predictor of increased mortality (HR = 4.1; 95% CI: 1.2–14; *p* = 0.025), while multifocality predicted both mortality (HR = 2.3; 95% CI: 1.3–3.9; *p* = 0.003) and reduced time to progression (HR = 3.3; 95% CI: 1.02–10.6; *p* = 0.04). **Conclusions**: This study underscores the importance of sex and molecular profiling in GB management, revealing distinct patterns in tumor characteristics and treatment administration between males and females. Our findings advocate for the integration of sex-specific considerations and molecular profiling into clinical decision-making to improve outcomes for GB patients.

## 1. Introduction

Sex differences are reported in the biology and the epidemiology of many tumors. Across all tissue types, cancer shows an overall male prevalence. Sex-specific risk factors, sex hormones, genetic and immunological mechanisms may play a role [1,2]. In particular the male-to-female ratio is higher in some primary brain tumors (such as malignant gliomas, medulloblastomas, and ependymomas) both in the adult and pediatric age, while meningiomas are more prevalent in females [1].

Glioblastoma (GB) is the most common primary malignant brain tumor, arising from glial cells; it has an average annual age-adjusted incidence rate of 3.21/100,000 population [3]. The exact mechanisms of glioblastoma oncogenesis, including the identification of the glioma-initiating cell, are yet to be discovered. While traditionally considered to arise from glial cells, emerging evidence suggests that glioblastomas may also originate from neural stem/progenitor cells or other cell types [4].

The incidence of GB is 1.6 times higher in males, while it is nearly similar between sexes for low-grade gliomas [5]. In both human and animal studies, females show a longer survival [3]. The reasons behind these differences are still unclear and many hypotheses have been brought up concerning the role of endocrine and immune system, molecular and genetic mechanisms [1,5]. The 2021 WHO CNS tumor classification [6] integrated the histopathological features of the glial tumors with the molecular profiles to provide a more concise diagnosis. This brought a rising interest towards GB molecular biomarkers, also, in the field of sex-differences, in regards to outcome and treatment response.

According to the 2021 WHO classification [6], the molecular biomarkers characterizing a GB are isocitrate dehydrogenase ½ (*IDH 1-2*) status, epidermal growth factor receptor (*EGFR*) amplification, telomerase reverse transcriptase (*TERT*) gene mutation, alpha-thalassemia/mental retardation, X-linked protein (*ATRX*) loss, and O6-methylguanine DNA methyltransferase (*MGMT*) promoter methylation [6].

*MGMT* promoter methylation, *IDH1*, and *ATRX* mutations are known as positive prognostic biomarkers [7,8]. On the other side *EGFR* amplification and overexpression and *TERT* promoter mutation are considered negative predictive biomarkers [9,10]. Many studies show that, in gliomas as in other tumors, the amplification of *EGFR* represents a risk factor for cell proliferation and could potentially interact with sex steroids [11,12]. A female-specific association with *TERT* expression seems also to account for GB risk [13]. Furthermore, *MGMT* promoter methylation was more commonly found in females with GB, with a better outcome after treatment with alkylating agent temozolomide (TMZ) [14,15]. A small series showed that GBs in female patients were *ATRX*-positive in over 90% of cases [16].

Lower incidence, longer time to progression, and overall survival in female patients affected by high-grade gliomas have been observed but the reasons for these disparities between sexes have not been cleared yet.

The aim of our study was to stratify clinical and biomolecular characteristics of GB by sex in a single-center cohort and to correlate them to treatment outcome.

## 2. Materials and Methods

### 2.1. Study Design

A single-center retrospective analysis was conducted using data from our institution’s Neuro-oncological Multidisciplinary Meeting (MDM) database, covering the period between 2018 and 2022. Our hospital serves a population of approximately 350.000 inhabitants and is the sole provider of neuro-oncological care in the region. Diagnoses were made based on radiological and histological/biomolecular criteria, according to the WHO 2021 classification [17].

Consecutive adult patients (≥18 years old) with a diagnosis of primary brain tumor, discussed within the neuro-oncological MDM and who signed a written informed consent, were included in this study.

The primary objective of this study was to assess and stratify the collected variables by sex and age, with a focus on identifying potential differences in tumor characteristics, molecular profiles, treatment patterns, and clinical outcomes between males and females diagnosed with a GB. The secondary objective was to identify independent predictors of time to progression and overall mortality in this sub-cohort of patients. Specifically, we aimed to evaluate the influence of age, sex, molecular biomarkers, and treatment modalities on these outcomes.

Tumor progression was defined according to standardized criteria [18], while overall mortality was calculated from the date of diagnosis to the date of death or last follow-up. Follow-up was defined by the latest clinical visit at the time of collection of the data.

Clinical data were collected by two of the authors between October 2020 and December 2023 and included in an anonymized database available only to the study researchers.

For each patient the following variables were collected and analyzed: sex, age, type of tumor (WHO classification 2021) [6], molecular biomarkers (IDH mutation status, *MGMT* promoter methylation, 1p/19q co-deletion, EGFR amplification, ATRX expression, and TERT mutation status), treatment modalities (surgery, chemotherapy, radiotherapy, and tumor treating fields (TTF)).

The results of this research are reported in accordance with the Strengthening the Reporting of Observational Studies in Epidemiology (STROBE) reporting guidelines [19].

This study was approved by the local ethical committee.

### 2.2. Statistical Analysis

We presented quantitative data as mean with standard deviation (SD) and qualitative data as absolute numbers with the corresponding percentages. We compared the different variables according to sex with the Student *t*-test, the Mann–Whitney test, the chi-squared test, or the Fisher’s exact test as appropriate. Missing data were not imputed. We performed a time-to-event analysis in order to assess differences in the occurrence of death and the occurrence of tumor progression. To assess the effects of different predictor variables on the two above-mentioned time-to-event outcomes, we first performed a univariate Cox proportional hazards regression, followed by a multivariable Cox regression model. Variables in the models by the authors were preselected based on their clinical relevance. We assessed the proportional hazard assumption of the Cox model by using the Schoenfeld test. For all regression models, we presented hazard ratios (HRs) and the corresponding 95% confidence intervals (95%-CIs). We performed all statistical tests two-sided, and a *p*-value < 0.05 was considered statistically significant. We used the Stata version 17.0 software (StataCorp LP, College Station, TX, USA) for all statistical analyses.

## 3. Results

We included 189 patients with primary glial brain tumors (F: 43%; M: 57%; *p* = 0.18), comprising 125 GB (F: 39%; M: 61% M; *p* = 0.17) 48 astrocytomas (F: 46%; M: 54%; *p* = 0.74), and 16 oligodendrogliomas (F: 62.5%; M: 37.5%; *p* = 0.1). The median age of the entire cohort was 61 years (IQR 43–71), with no significant difference between sexes (F: 60 years, IQR 45–71; M: 61 years, IQR 42–71; *p* = 0.55). Among GB patients, the median age was 65 years (IQR 56–73), also without significant sex-based differences (F: 67 years, IQR 57–74; M: 64 years, IQR 55–73; *p* = 0.44) (Table 1).

The results show a significantly higher rate of multifocality at presentation and EGFR amplification in female patients compared to males. Conversely, the administration of radiotherapy and chemotherapy was significantly more prevalent among male patients.

Subsequent analyses focused exclusively on GB patients. Tumor location and laterality at diagnosis did not differ significantly between sexes. However, multifocality at presentation was observed in 17.5% of patients, with a significantly higher prevalence in females (F: 14%; M: 8%; *p* = 0.01).

No significant sex-based differences were observed in *MGMT* promoter methylation (F: 62%; M: 72%; *p* = 0.27), *ATRX* expression (F: 100%; M: 87%; *p* = 0.08), or *TERT* mutation status (F: 84%; M: 79%; *p* = 0.6). In contrast, *EGFR* amplification was significantly less frequent in males compared to females (F: 52.5%; M: 25.5%; *p* = 0.007).

Chemotherapy and radiotherapy were administered more frequently to males than females; chemotherapy was given to 63% of females versus 80% of males (*p* = 0.04), and radiotherapy was administered to 67% of females versus 84% of males (*p* = 0.03). A similar, though non-significant, trend was observed for TTF usage (F: 10%; M: 14.5%; *p* = 0.48).

Most patients underwent surgical intervention, with no significant differences between sexes (F: 94%; M: 96%; *p* = 0.58).

No significant sex-based differences were observed in overall survival (Figure 1A). However, survival was positively associated with *MGMT* promoter methylation (*p* = 0.002) (Figure 1B). No significant differences in survival time according to *MGMT* promoter methylation status were observed between female and male patients (Figure 1B,C).

Survival was negatively associated with TERT mutation (*p* = 0.01) in both sexes.

Multivariable analysis of molecular biomarkers (*MGMT*, *EGFR*, *ATRX*, and *TERT*) stratified by sex revealed a significant increase in mortality among all patients with *TERT* promoter mutation (*p* = 0.025; HR = 4.1; 95% CI: 1.2–14). A second multivariable model, incorporating age at diagnosis, sex, multifocality, and treatment modalities, identified multifocality as a significant predictor of mortality (HR = 2.3, 95% CI: 1.3–3.9; *p* = 0.003). Additionally, multifocality was associated with reduced time to progression (HR = 3.3, 95% CI: 1.02–10.6; *p* = 0.04).

## 4. Discussion

Our study, aiming at assessing the epidemiological and molecular biomarker characteristics of our cohort, confirms that GB is the most frequent type of glioma and that it is more frequent in males. The prevailing hypothesis for this sex disparity involves hormonal mechanisms, particularly the role of androgen receptor signaling. Specifically, androgen receptor overexpression has been linked to enhanced tumor growth and invasive potential in GB [20].

In line with previous studies on larger cohorts, we found a male/female ratio of about 1.6:1 [21].

Sex was defined based on biological criteria. We considered sex as what is related to the karyotype and gonads and gender as what is related to the social, behavioral, and cultural aspects of being feminine or masculine [22].

Sex and gender were always coincident in our cohort of patients; therefore, we could not infer any considerations about gender and sex interaction and cancer in our study [22].

Regarding sex-based differences we found that, while multifocality at diagnosis was found in 17% of the patients, females (although known to exert a better survival [20]), showed a significantly higher number of multifocal tumors at presentation. In the literature, this percentage varies from 2 to 35% of the cases [23] but there are no specific data regarding sex distribution. Although speculative, these data may go together with the female prevalence of *EGFR* amplification that was also found in our cohort. As a matter of fact, preclinical studies show that *EGFR* amplification may drive the non-angiogenetic invasivity of these tumors [24,25]. Estradiol has been shown to directly promote proliferation, migration, and invasiveness in glioblastoma cells, as demonstrated by in vitro studies [26]. We might hypothesize that the slower initial disease progression observed in female patients could be attributed to a more robust hormonally modulated immune response against the tumor [27] or potential delays in medical referral among women [22]. In agreement with a previous report, we also observed that *ATRX* was expressed by immunohistochemistry in >90% of female subjects [16]. *TERT* mutation was associated with a higher mortality in both sexes, while male sex and older age at diagnosis with a faster progression.

We did not find differences between sexes in *MGMT* methylation promoter status, as described in a previous report [28], but we confirmed it as a positive prognostic marker on survival on both sexes. Likely due to limited statistical power from our small sample size, we did not detect a significant survival advantage for female patients with *MGMT*-methylated tumors, as was already shown by Barnett et al. [15].

Regarding treatments, we observed that female patients received fewer treatments (chemotherapy and radiotherapy) than males. This finding seems to be independent from the severity of the disease at presentation. Previous reports already brought to light disparities between sex concerning medical treatments and cancer treatment [29,30,31,32,33,34,35], which hinder females. Female patients for instance are less likely to be treated for heart disease or for irritable bowel syndrome and receive less imaging and more antidepressants [34,35,36]. The reason for this disparity, which is not limited to low-income countries, has still to be clarified. It should be noted that a comprehensive database analysis conducted in the United States on 124,502 of GB also revealed that women received fewer treatments overall, including radiation therapy and surgery, for both GB and low-grade glioma [37]. Notwithstanding, another wide registry study carried out in 2018 in the USA observed that female patients, although presenting the same type and extension of GB, achieved an independent sex-based better cancer-specific survival [21]. Interestingly another study on the same registry showed that being married leads to better outcomes than single status, independently from the type of treatments [38]; unfortunately, we were not able to assess the marital status of the our participants, together with other social and gender-related variables. We did not find a significant difference in the diagnosis of tumor-related epilepsy in our population but a trend toward more male participants being affected. In this regard, a prior study found that the prevalence of seizures was twice as high in male patients, while epilepsy was more likely to occur in female patients with left-sided tumors [39].

## 5. Conclusions

Despite the clear limitations of our study (due to its retrospective design and the limited sample size), we observed that our cohort reflects the already published characteristics and distribution of gliomas. It also has taken into account some of the novel clinical molecular diagnostic biomarkers of GB together with the data on treatments and survival studied as sex-related variables. In fact GB is one of the tumors where there are well-known sex differences, but the reasons for this tumor dimorphism are not known yet. Our results may give a contribution in this direction. The prognosis of these types of tumors is indeed still severe, and better knowledge on pathophysiological and biological mechanisms underlying the sex differences in incidence and survival may shed light on possible future therapeutical targets.

Additionally, we believe that social factors may still be at play in the lower access to therapies observed in female participants compared to males in our cohort. This has to be brought to the attention of the medical community and debated.

## Figures and Tables

**Figure 1 biomedicines-13-01715-f001:**
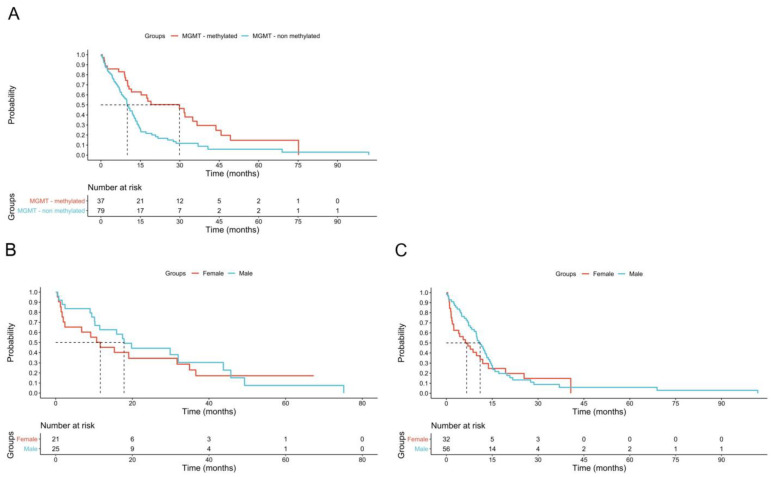
Effect of MGMT promoter methylation on in females and males. (**A**): effect of MGMT methylation on survival of all patients (median survival in MGMT + 29.86 months vs MGMT–10.86 months *p* = 0.001). (**B**): the Kaplan Mayer curve of MGMT + subjects shows no difference on survival for female and male (median survival female 11.83 months vs male 17.83 *p* = 0.5). (**C**): the Kaplan Mayer curve of MGMT–subjects shows no difference on survival for female and male subjects (median survival female 6.63 months vs male 11.03 *p* = 0.3).

**Table 1 biomedicines-13-01715-t001:** Clinical and molecular characteristics of the study cohort: overall population and glioblastoma stratified by sex.

	Female (%)	Male (%)	All (%)	p Value
Age, years (IQR)	60 (45–71)	61 (42–71)	61 (43–71)	0.55
Gliomas	81 (43)	108 (57)	189	0.18
Glioblastoma	49 (39)	76 (61)	125	0.17
Astrocytoma	22 (46)	26 (54)	48	0.74
Oligodendroglioma	10 (62.5)	6 (37.5)	16	0.12
Glioblastoma				
Multifocality	14 (28.5)	8 (10.5)	22 (17.5)	0.01
Molecular profile				
IDH1 wt	45 (98)	70 (97)	115 (97.5)	0.83
MGMT unmethylated	28 (62)	51 (72)	69 (68)	0.28
TERT mutation	26 (84)	34 (79)	60 (81)	0.6
EGFR amplification	20 (52.5)	14 (25.5)	34 (36.5)	<0.01
ATRX expression	22 (100)	42 (87)	64 (91.5)	0.08
Chemotherapy	31 (63)	60 (80)	91 (73.5)	0.04
Radioteraphy	33 (67)	63 (84)	96 (77.5)	0.03
TTF	5 (10)	11 (14.5)	16 (13)	0.48
Biopsies	15 (33.5)	14 (19.5)	29 (25)	0.2
Surgery	46 (94)	73 (96)	119 (95)	0.58
Re-surgery	7 (17)	14 (23)	21 (20.5)	0.5
Epilepsy	19 (39.5)	37 (50)	56 (46)	0.15
MRI progression	28 (61)	51 (71)	79 (67)	0.25
Mortality	38 (78)	65 (85)	103 (82.5)	0.25

IQR (interquartile range); age is expressed as median values; IDH (isocitrate dehydrogenase); *MGMT* (O-6-methylguanine-DNA methyltransferase); TERT (telomerase reverse transcriptase); EGFR (epidermal growth factor); ATRX (alpha-thalassemia/mental retardation X-linked); TTF (tumor treating fields); MRI (magnetic resonance imaging).

## Data Availability

The data presented in this study are available on request from the corresponding author due to privacy reasons.

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
