# Peer review of "Sex-Related Differences in Glioblastoma: A Single-Center Retrospective Cohort Study"

_biomedicines, 2025, doi:10.3390/biomedicines13071715_

Round 1
Reviewer 1 Report
Comments and Suggestions for Authors
In this manuscript (3678760), Chiara et al. undertake a retrospective study on a single center cohort to assess sex differences in glioblastoma (GBM). Given the interest in sex differences in GBM, this study confirms many previous findings and provides some novel observations, including sex differences in receiving treatment (males more frequent than females), multi-focal disease (females more frequent than males), and EGFR amplification (males more frequent than females). Taken together, while this is still a single center study, it provides some new information that will likely benefit the field. In general, my enthusiasm for this manuscript is high, but some additional clarification is needed prior to publication.
- In the abstract, access to care is mentioned (line 19) but given this is not being studied, it should be removed and if the authors are interested in this topic, it can be explored in the discussion.
- In the abstract, for consistency, I would recommend stating which sex has the higher phenotype/genotype, line 28, it would read better if stated, EGFR amplification was more frequent in males.
- In line 60, the IDH1/2 has become a superscript, this is a minor typo and should be fixed.
- It is nice to see sex and gender being concordant (line 107), but I would recommend focusing on sex here and this mention can go in the discussion. I would also recommend an expansion in the discussion about sex versus gender as these two terms are quite different but often are used interchangeably. I would also take this opportunity to cite a recent review on this topic as well as sex/gender interactions (PMID 40389544).
- The sex difference in GBM incidence (line 129) is consistent with previous reports but it is unclear how this large difference is not statistically significant.
- It would be useful to graph the sex difference in survival with a clear indication of median survival. It would also be worth doing this in the context of MGMT methylation.
- In the discussion, it would be important to include sex differences in survival based on MGMT methylation (PMID 38611052) as it is mentioned that there is no sex difference in MGMT methylation (reference 22, line 177) but the sex difference may not be in the incidence of MGMT methylation but survival.
Author Response
Dear Reviewer,
Thank you for your time and for your thoughtful and pertinent comments. Please find our responses below.
- In the abstract, access to care is mentioned (line 19) but given this is not being studied, it should be removed and if the authors are interested in this topic, it can be explored in the discussion. R: Thank you for your remark, we revised the sentence (see line 19)
- In the abstract, for consistency, I would recommend stating which sex has the higher phenotype/genotype, line 28, it would read better if stated, EGFR amplification was more frequent in males. R: We agree with your suggestion and rephrased the sentence as asked (see line 27-28).
- In line 60, the IDH1/2 has become a superscript, this is a minor typo and should be fixed. R: the superscript has been adjusted.
- It is nice to see sex and gender being concordant (line 107), but I would recommend focusing on sex here and this mention can go in the discussion. I would also recommend an expansion in the discussion about sex versus gender as these two terms are quite different but often are used interchangeably. I would also take this opportunity to cite a recent review on this topic as well as sex/gender interactions (PMID 40389544). R: Thank you very much for this comment and for the reference regarding gender and sex interaction in cancer, which we have been happy to include and cite. Please find our additions in the discussion (lines 183-187).
- The sex difference in GBM incidence (line 129) is consistent with previous reports but it is unclear how this large difference is not statistically significant. R: : Thank you for this observation who let us modify this analysis in order to make it clearer. The p values ​​reported for histology were based on a three-group comparison, but we have now modified this to a two-group comparison (GB vs non-GB; astrocytoma vs non-astrocytoma, oligodendroglioma vs non-oligodendroglioma) so that it is clearer, although the difference is still not significant (likely due to the sample size). We also decided to switch to the median values instead of the mean values. We modified the values witin the table and in the text (see lines 131-137).
- It would be useful to graph the sex difference in survival with a clear indication of median survival. It would also be worth doing this in the context of MGMT methylation. R: We appreciated this suggestion and provided a figure containing the graphs of sex differences in survival (Fig1a), and MGMT methylation in male and females (Fig 1b&c)
- In the discussion, it would be important to include sex differences in survival based on MGMT methylation (PMID 38611052) as it is mentioned that there is no sex difference in MGMT methylation (reference 22, line 177) but the sex difference may not be in the incidence of MGMT methylation but survival. R: We appreciated very much your comment, which led us to reconsider also the survival improvement due to MGMT promoter methylation pattern in both female and male subjects. Although we confirm the advantage of the MGMT methylation on survival, we could not detect a significant differences between sex or within a specific sex compared to the other (as shown by Barnet A et al). See lines 163-166.
Please see attachment (revised file).
Reviewer 2 Report
Comments and Suggestions for Authors
This is an interesting cohort study about sex differences in gliomas, with a particular focus on glioblastomas in Switzerland. However, several points require clarification and additional detail to enhance the rigor and completeness of the manuscript.
1. Throughout the abstract and the main text, glioblastoma is referred to as “GBM.” However, according to the most recent WHO classification of CNS tumors (2021), the term "glioblastoma" should be abbreviated as “GB” rather than “GBM,” since "multiforme" is no longer part of the nomenclature. Please modify the abbreviation throughout the manuscript.
2. Please give a current explanation of glioblastoma origins. While traditionally considered to arise from glial cells, recent evidence suggests that glioblastomas may also originate from neural stem/progenitor cells or other cell types.
3. Please include a clear and complete legend of the Table. All abbreviations used in the table should be defined, and statistically significant differences should be clearly described.
4. It is necessary to include a discussion of the potential mechanisms underlying the observed sex differences, particularly with regard to multifocal tumor presentation. It is recommended to include a brief discussion of the possible genetic, molecular, hormonal or microenvironmental factors contributing to these differences.
5. Given the well-established influence of sex steroid hormones on glioblastomas progression, the authors should consider including a discussion of how these may relate to the sex-specific findings reported in the study. I recommend referencing and discussing the following review articles and consulting the primary literature they cite to support this addition:
1. Alemán OR, Quintero JC and Camacho-Arroyo I. The language of glioblastoma: a tale of cytokines and sex hormones communication. Neuro-Oncology Advances. 2025. 7 (1). Doi: 10.1093/noajnl/vdaf017.
Author Response
Dear Reviewer,
thank you for taking time to review our paper and for your precious comments.
- Throughout the abstract and the main text, glioblastoma is referred to as “GBM.” However, according to the most recent WHO classification of CNS tumors (2021), the term "glioblastoma" should be abbreviated as “GB” rather than “GBM,” since "multiforme" is no longer part of the nomenclature. Please modify the abbreviation throughout the manuscript. Thank you for pointing out this error, the text has been revised accordingly to your observation.
- Please give a current explanation of glioblastoma origins. While traditionally considered to arise from glial cells, recent evidence suggests that glioblastomas may also originate from neural stem/progenitor cells or other cell types. Thank you for this suggestion, we added explanations and a reference in line with your observation (lines 50 to 54, ref 4).
- Please include a clear and complete legend of the Table. All abbreviations used in the table should be defined, and statistically significant differences should be clearly described. We agree with your kind remark and added the legend to table 1
- & 5 It is necessary to include a discussion of the potential mechanisms underlying the observed sex differences, particularly with regard to multifocal tumor presentation. It is recommended to include a brief discussion of the possible genetic, molecular, hormonal or microenvironmental factors contributing to these differences. Given the well-established influence of sex steroid hormones on glioblastomas progression, the authors should consider including a discussion of how these may relate to the sex-specific findings reported in the study. I recommend referencing and discussing the following review articles and consulting the primary literature they cite to support this addition. Thank you for these comments, we completely agree with your observations and were happy to include the suggested reference. Please find our additions in lines 177-180 and 193-197.
Round 2
Reviewer 1 Report
Comments and Suggestions for Authors
In general, this is a well revised manuscript and likely a good candidate for publication in the current form, with a few minor edits.
1. The claim that "Notably, mesenchymal stromal cells have been implicated as potential glioblastoma-initiating cells. (4). (lines 53, 54)" should be removed as this is a single paper in this area and there are many more on neural progenitor cells that are not included. As this is in the introduction, the point can be made without including this paper.
2. I would also recommend using the abbreviation "GBM" for glioblastoma as it is still commonly used in the field.
Author Response
Dear Reviewer,
thank you for your comments.
As suggested we removed lines 53 & 54.
Regarding the abbreviation GBM, in the first round of revision, reviwer #2 required changing GBM into GB "Throughout the abstract and the main text, glioblastoma is referred to as “GBM” however, according to the most recent WHO classification of CNS tumors (2021), the term "glioblastoma" should be abbreviated as “GB” rather than “GBM,” since "multiforme" is no longer part of the nomenclature. Please modify the abbreviation throughout the manuscript".
Therefore the manuscript was edited according to this observation.
We will be happy to modify the text in accordance with what is requested, we defer to the editor's decision in this regard.